# The long non-coding RNA *Dali* is an epigenetic regulator of neural differentiation

Vladislava Chalei[1], Stephen N Sansom[1,2], Lesheng Kong[1], Sheena Lee[3], Juan F Montiel[1], Keith W Vance[1*†‡], Chris P Ponting[1*†]

[1]MRC Functional Genomics Unit, Department of Physiology, Anatomy and Genetics, University of Oxford, Oxford, United Kingdom; [2]Computational Genomics Analysis and Training Programme, University of Oxford, Oxford, United Kingdom; [3]Department of Physiology, Anatomy and Genetics, University of Oxford, Oxford, United Kingdom

**Abstract** Many intergenic long noncoding RNA (lncRNA) loci regulate the expression of adjacent protein coding genes. Less clear is whether intergenic lncRNAs commonly regulate transcription by modulating chromatin at genomically distant loci. Here, we report both genomically local and distal RNA-dependent roles of *Dali*, a conserved central nervous system expressed intergenic lncRNA. *Dali* is transcribed downstream of the *Pou3f3* transcription factor gene and its depletion disrupts the differentiation of neuroblastoma cells. Locally, *Dali* transcript regulates transcription of the *Pou3f3* locus. Distally, it preferentially targets active promoters and regulates expression of neural differentiation genes, in part through physical association with the POU3F3 protein. *Dali* interacts with the DNMT1 DNA methyltransferase in mouse and human and regulates DNA methylation status of CpG island-associated promoters in *trans*. These results demonstrate, for the first time, that a single intergenic lncRNA controls the activity and methylation of genomically distal regulatory elements to modulate large-scale transcriptional programmes.

**\*For correspondence:**
k.w.vance@bath.ac.uk (KWV);
Chris.Ponting@dpag.ox.ac.uk (CPP)

†These authors contributed equally to this work

Present address: ‡Department of Biology and Biochemistry, University of Bath, Bath, United Kingdom

**Reviewing editor**: Thomas R Gingeras, Cold Spring Harbor Laboratory, United States

## Introduction

A growing number of nuclear localised long noncoding RNAs (lncRNA, ≥ 200 nt) are known to regulate gene transcription and chromatin organisation (reviewed in (*Vance and Ponting, 2014*)). Many of these transcripts appear to act near to their site of synthesis to regulate the expression of genes locally on the same chromosome (*cis*-acting). *Cis*-acting lncRNA regulatory mechanisms have been described in detail for a number of enhancer associated nuclear lncRNAs, as well as lncRNAs involved in the processes of genomic imprinting and X chromosome inactivation (*Tian et al., 2010*; *Melo et al., 2013*; *Monnier et al., 2013*; *Mousavi et al., 2013*; *Santoro et al., 2013*; *Vallot et al., 2013*). Some *cis*-acting lncRNAs bind to DNA methyltransferase (DNMT) proteins and regulate genomic DNA methylation levels specifically at their sites of transcription (*Mohammad et al., 2010*; *Di Ruscio et al., 2013*).

*Trans*-acting lncRNAs that regulate gene expression across multiple chromosomes and on either allele have been documented less frequently. The ability of such lncRNAs to exert widespread effects on gene expression *in trans* is poorly understood, in large part because direct transcriptional targets for only very few of these transcripts have thus far been identified (*Chu et al., 2011*; *Ng et al., 2013*; *Simon et al., 2011*; *Vance et al., 2014*). Moreover, it is not clear whether these transcripts commonly act directly, or within ribonucleoprotein complexes, and how they might modify their target genes' regulatory landscape such as by regulating their DNA methylation profiles.

**eLife digest** Traditionally genes are considered to contain all the instructions necessary to build proteins. For these instructions to be followed they need to be 'transcribed' into molecules called messenger RNA, which are then 'translated' to form the protein. Messenger RNAs are not the only type of RNA molecule made in a cell; long non-coding RNAs (or lncRNAs), for example, are transcribed but never translated into proteins. Instead, some lncRNAs control the expression of nearby genes and some alter how the DNA is packaged within the cell.

Several lncRNAs have been found to control their neighbouring genes, but it is unclear how many of these molecules can also regulate genes that are much further away, even on other chromosomes. One lncRNA called *Dali* is made in cells of the nervous system of mammals. In the genome, the gene for *Dali* is situated next to a gene called *Pou3f3,* which encodes a protein that contributes to the growth and development of nerves and the kidneys.

Chalei et al. have now shown that artificially reducing the amount of the *Dali* lncRNA restricts the development of mouse cells called N2A cells, which are commonly used to study the development of nerve cells. Reducing *Dali* lncRNA levels in these cells caused *Pou3f3* messenger RNA levels to also decrease, which demonstrates that *Dali* is a lncRNA that controls its neighbouring gene. The levels of many other genes were also changed when *Dali* levels were reduced, including many genes that are needed to grow working nerve cells.

Chalei et al. also showed that the *Dali* lncRNA binds to 1427 different regions of the genome of N2A cells, most often near to the start of active genes; *Dali* could be carried to these sites by the POU3F3 protein. The DNA sequences with which the *Dali* lncRNA binds were all different. Chalei et al. found that *Dali* also binds to an enzyme, called DNMT1, that chemically modifies DNA to change how it is packaged into a cell, and they predict that this enzyme helps *Dali* to find its binding sites. Furthermore, when *Dali* lncRNA levels were artificially reduced, the chemical modifications that affect the packaging of DNA in the cell—and hence the expression of genes encoded by this DNA—were changed for several genes. Some of these genes were located far away from the gene that encodes *Dali*, indicating that this lncRNA can regulate the packaging and expression of distant genes.

Many genes that are regulated by *Dali* are also regulated by the POU3F3 protein; this suggests that the lncRNA might work together with this protein to affect the expression of some genes. Further work is now needed to uncover how many other lncRNAs act away from their sites of synthesis, and how many also form complexes with DNA-binding and DNA-modifying proteins.

Many thousand mammalian intergenic lncRNAs have now been identified. Not all lncRNA transcript models will be functional, however. Single exon models, in particular, can be artefacts arising from genomic DNA contaminating sequencing libraries, and transcripts that are expressed at average levels lower than one copy per cell are less likely to confer function. Highly and broadly expressed, and bona fide monoexonic intergenic lncRNAs, such as *Neat1* and *Malat1/Neat2*, however, appear not to have essential roles because their knockout mouse models are viable and fertile (*Eissmann et al., 2012*; *Zhang et al., 2012*). Transcript sequences and levels are thus not reliable predictors of mechanism. Instead, the significant temporal and spatial co-expression of genomically adjacent intergenic lncRNA and transcription factor genes might suggest that such lncRNAs commonly modulate transcriptional programmes that are initiated by these transcription factors (*Ponjavic et al., 2009*). Indeed, several intergenic lncRNAs have well-documented *cis*-acting regulatory roles (*Wang et al., 2011*; *Zhang et al., 2012*; *Berghoff et al., 2013*).

Spatiotemporal co-expression of intergenic lncRNA and transcription factor genes is most pronounced during the development of the mouse central nervous system (CNS) (*Ponjavic et al., 2009*). To investigate the mechanistic basis of this physical linkage we chose to study a 3.5-kb, CNS-expressed, monoexonic, intergenic lncRNA termed *Dali* (DNMT1-Associated Long Intergenic), owing to its conservation of sequence and transcription across therian mammals and its genomic proximity to a transcription factor gene, *Pou3f3* (also known as *Brn1* or *Oct8*), which encodes a class III POU family transcription factor. *Dali* is transcribed in the sense orientation, relative to *Pou3f3*, from a locus 50 kb

downstream of *Pou3f3* within the flank of an extended genomic region (*Figure 1A*) that is characterised by near pervasive transcription in neuronal lineages (*Ramos et al., 2013*). Sauvageau et al. recently generated mouse knockout models for two of these intergenic lncRNA loci, *linc-Brn1a*, and *linc-Brn1b* (*Figure 1A*). Genomic deletion of the *linc-Brn1b* locus resulted in significant (~50%) down-regulation of the upstream *Pou3f3* gene, and *linc-Brn1b*[-/-] mice exhibited abnormalities of cortical lamination and barrel cortex organization (*Sauvageau et al., 2013*). These abnormalities may derive from loss of the *linc-Brn1b* RNA transcript, or from the deletion of DNA functional elements (*Bassett et al., 2014*). The *Dali* locus is more distally located and does not overlap previously described lncRNA loci or regulatory elements (*Figure 1A*).

*Pou3f3* is a single exon gene whose protein binds to DNA in a sequence-specific manner. *Pou3f3* contributes to both neuronal and kidney development by regulating the proliferation and differentiation of progenitor cells (*Nakai et al., 2003*). Mouse mutants with homozygous loss of *Pou3f3* die of renal failure within 36 hr *post partum* (*Nakai et al., 2003*), with severe defects of the hippocampus and forebrain among others (*McEvilly et al., 2002*). In the developing neocortex, *Pou3f3* is expressed in late neuronal precursors and in migrating neurons and, together with its closely related paralogue *Pou3f2*, is required in ventricular zone progenitors for deep-to-upper layer fate transition, sustained neurogenesis and cell migration (*Dominguez et al., 2013*).

Our experiments show that *Dali* is required for the normal differentiation of neural cells in culture. Furthermore, our results indicate that *Dali* functions by modulating the expression of its neighbouring *Pou3f3* gene, as well as by interacting with the POU3F3 protein, and by directly binding and regulating the expression of genes involved in the neuronal differentiation programme *in trans*. Unexpectedly, *Dali* associates with the DNMT1 DNA methyltransferase and reduction of *Dali* levels increases DNA methylation at a subset of *Dali*-bound and -regulated promoters *in trans*. Our data therefore provide the first evidence that a lncRNA transcript can regulate multiple genes situated away from its site of synthesis by binding to promoter-proximal regulatory elements and altering their DNA methylation status in *trans*.

## Results

### Conserved *Dali* genomic organisation and transcription

Full-length mouse *Dali* is approximately 500 nt (2.6 kb) longer than a previously identified *AK034039* cDNA cloned from the telencephalon (*Figure 1—figure supplement 1A*). Its locus, downstream of the *Pou3f3* gene, contains mammalian conserved sequence both just upstream of its transcriptional start site, which presumably contributes to this locus' promoter, and within its transcribed sequence. A positionally equivalent and sequence-similar human *DALI* (~3.7 kb) transcript was identified by RT-PCR and RACE in human foetal brain (*Figure 1B*; *Figure 1—figure supplement 1B,C*). Transcriptional evidence also exists for the orthologous locus in rat embryonic, as well as heart and kidney, samples (data not shown).

### *Dali* is a chromatin-associated transcript that is co-expressed with *Pou3f3* in neural cell lineages

ENCODE data indicate that both mouse and human *Dali* loci have the properties of a weak (or poised) enhancer in both brain and kidney tissues (*Figure 1—figure supplement 1D,E*). Consistent with this, *Dali* was most highly expressed in the adult brain and kidney, two of the three tissues displaying highest *Pou3f3* expression, when profiled across a panel of adult mouse organs (*Figure 1—figure supplement 1F,G*). In adult mouse (P56), *Dali* and *Pou3f3* were expressed in all three regions of adult neurogenesis, the sub-ventricular zone (SVZ), olfactory bulb (OB), and dentate gyrus (DG) (*Figure 1—figure supplement 1I*) (Reviewed in *Ming and Song, 2011*). *Dali* was also co-expressed with *Pou3f3* temporally and spatially in the developing mouse embryonic brain (*Figure 1C,D*). Both transcripts were up-regulated with the first appearance of cortical neurons (E10.5), and increased in expression further as the ratio between neurons and progenitors grew (*Figure 1C,D*). Furthermore, both *Dali* and *Pou3f3* transcripts were undetectable in self-renewing mouse E14 embryonic stem (ES) cells, but after 3 days of retinoic acid (RA)-induced differentiation, a stage corresponding to the cell cycle exit of neuronal progenitors and their differentiation into neurons, these transcripts were rapidly up-regulated, their levels subsequently peaking at days 7 (*Pou3f3*) and 8 (*Dali*) (*Figure 1E,F*).

Mouse neuroblastoma N2A cells, which are frequently used as a neuronal progenitor-like cell type and an in vitro model of neuronal differentiation (*Tremblay et al., 2010*), express both *Dali* (at a

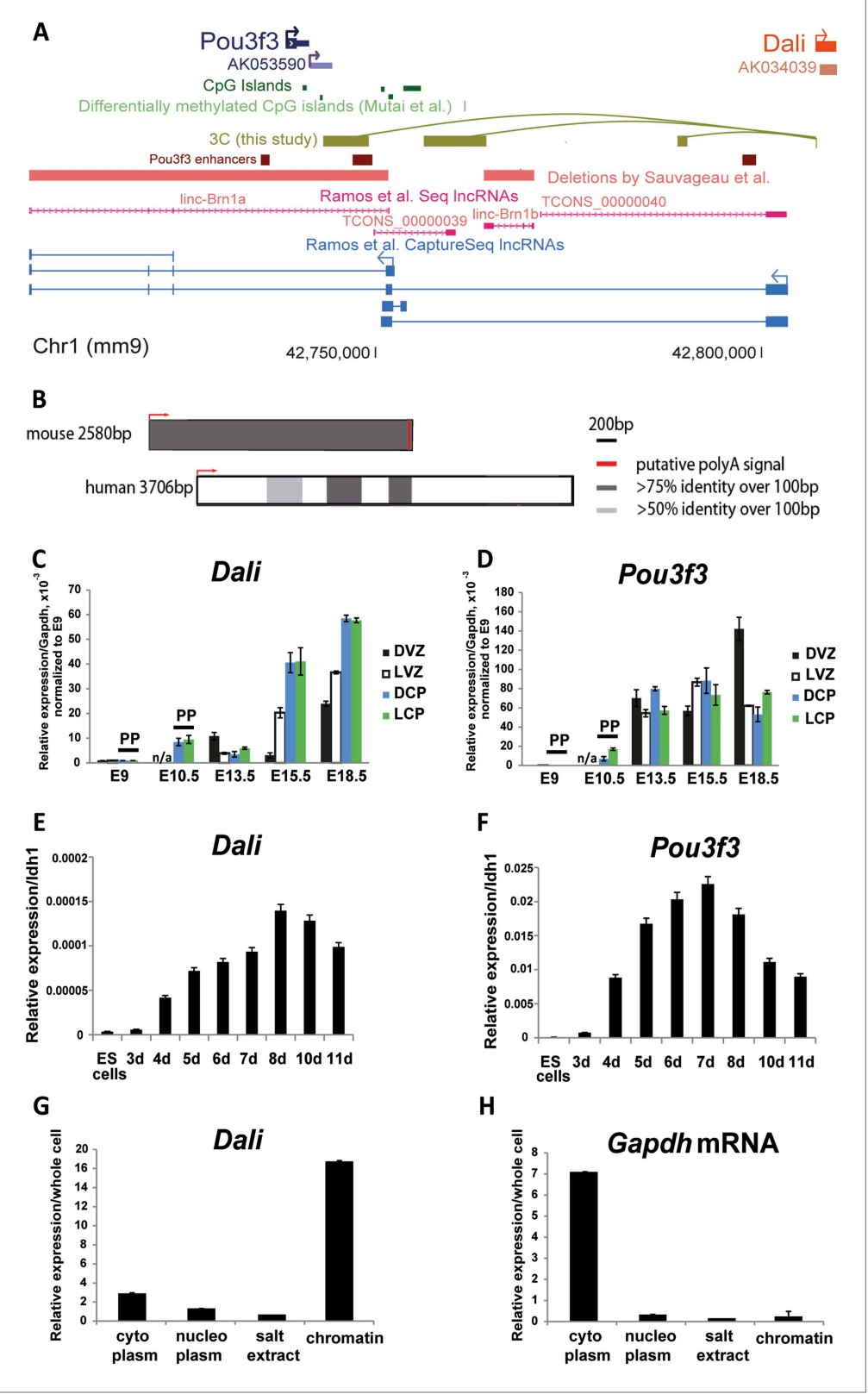

**Figure 1**. Conservation and expression within the *Dali* and *Pou3f3* loci. (**A**) Schematic illustration of the mouse *Pou3f3* genomic region showing coding and non-coding transcripts, enhancer elements from Vista Enhancer Browser (*Visel et al., 2007*), CpG islands, and published genomic deletions (*Sauvageau et al., 2013*). (**B**) Conservation and
*Figure 1. Continued on next page*

*Figure 1. Continued*

relative sizes of *Dali* transcripts in mouse and human confirmed by RACE. (**C**) *Dali* and (**D**) *Pou3f3* are co-expressed temporally and spatially in the developing mouse brain. DVZ: Dorsal ventricular zone; LVZ: Lateral ventricular zone; DCP: Dorsal cortical plate; LCP: Lateral cortical plate; PP: pre-plate. The levels of *Dali*, *Pou3f3* were measured by qRT-PCR. Results are normalised to *Gapdh* and presented relative to expression in E9.0 sample (set arbitrarily to 1). Mean ± s.e., n = 3 (technical replicates). (**E** and **F**) Similarly to *Pou3f3*, *Dali* is up-regulated during neuronal differentiation of mouse ES cells. Neuronal differentiation of mouse ES cells was induced using RA. The levels of *Dali* and *Pou3f3* were measured by qRT-PCR. Results are presented relative to an *Idh1* reference gene which does not change significantly during differentiation. Mean ± s.e., n = 3. (**G** and **H**) *Dali* is a chromatin associated transcript. The relative amounts of *Dali* (**G**) and a control mRNA (*Gapdh*) (**H**) in the indicated fractions were measured by qRT-PCR. Mean values ± s.e. of three independent experiments.

The following figure supplements are available for figure 1:

**Figure supplement 1**. Analysis of the mouse and human *Dali* loci.

**Figure supplement 2**. The Pou3f3 locus occurs in a folded nuclear conformation both prior to and after the onset of the expression of its transcripts.

population-average level of 2 copies per cell (*Figure 1—figure supplement 1K*)) and *Pou3f3*. When first detected in neuronal-progenitor-dominated areas of the developing brain (E10.5), *Dali* is expressed at a level at least two orders of magnitude higher than in N2A cells (*Figure 1—figure supplement 1H*). However, in N2A cells treated with RA for 72 hr, *Dali* is up-regulated approximately 4.5-fold, similar to the up-regulation observed in embryonic cortical plate (both dorsal and lateral) between days E10.5 to E18.5 (*Figure 1—figure supplement 1H*). Therefore, despite *Dali* expression level differences in N2A cells and the in vivo system, N2A cells represent an appropriate model system in which to study *Dali* function. Furthermore, *Dali*, but not a control mRNA (*Gapdh*), was highly enriched in the nucleus of N2A cells, most abundantly in the chromatin fraction (*Figure 1G,H*). Taken together, the data suggest that *Dali* may be involved in regulating nuclear function during neuronal development, potentially in coordination with *Pou3f3*.

## *Dali* regulates neural differentiation of N2A cells

We next investigated whether *Dali* regulates neural differentiation by generating three independent stable *Dali* knockdown N2A cell lines each showing approximately 50–70% reduction of *Dali* transcript levels and inducing neural differentiation using RA (*Figure 2A*). Compared to a stable non-targeting control line, fewer differentiated cells of *Dali* knockdown lines developed neurites. Those that did exhibited shorter neurites, often with multiple short outgrowths emanating from the same cell, compared to one or two long neurites developed by differentiated control cells (*Figure 2B,C*) indicating that *Dali* is required for normal differentiation of N2A cells.

## *Dali* regulates neural gene expression

To investigate the molecular function of *Dali*, we performed microarray analysis to profile the transcriptome of N2A cells in which *Dali* transcript levels had been depleted by ~70% using transient transfection of a specific *Dali* targeting shRNA expression vector (*Figure 2D*; shRNA and RT-qPCR oligo sequences and positions can be found in *Supplementary file 1*). *Dali* knockdown resulted in statistically significant changes in expression levels for 270 genes (False Discovery Rate [FDR] < 10%) compared to a non-targeting control (*Supplementary file 2*; *Figure 2E*). 14 of 15 of these genes were also determined as being differentially expressed, with similar fold changes, using RT-qPCR and two additional independent shRNA expression constructs targeting *Dali* (*Figure 2—figure supplement 1C*). Gene expression changes we observed using microarrays were thus unlikely to be dominated by off-target effects of the shRNA used. Gene Ontology (GO) analysis revealed that *Dali*-regulated genes were significantly enriched in cell cycle, DNA repair, cellular response to stimulus, and cell projection assembly functions (*Figure 2F* and *Supplementary file 2*; Benjamini-Hochberg corrected p ≤ 0.05). Taken together, these expression and loss of function studies suggest that *Dali* acts as a pro-differentiation factor in neural development.

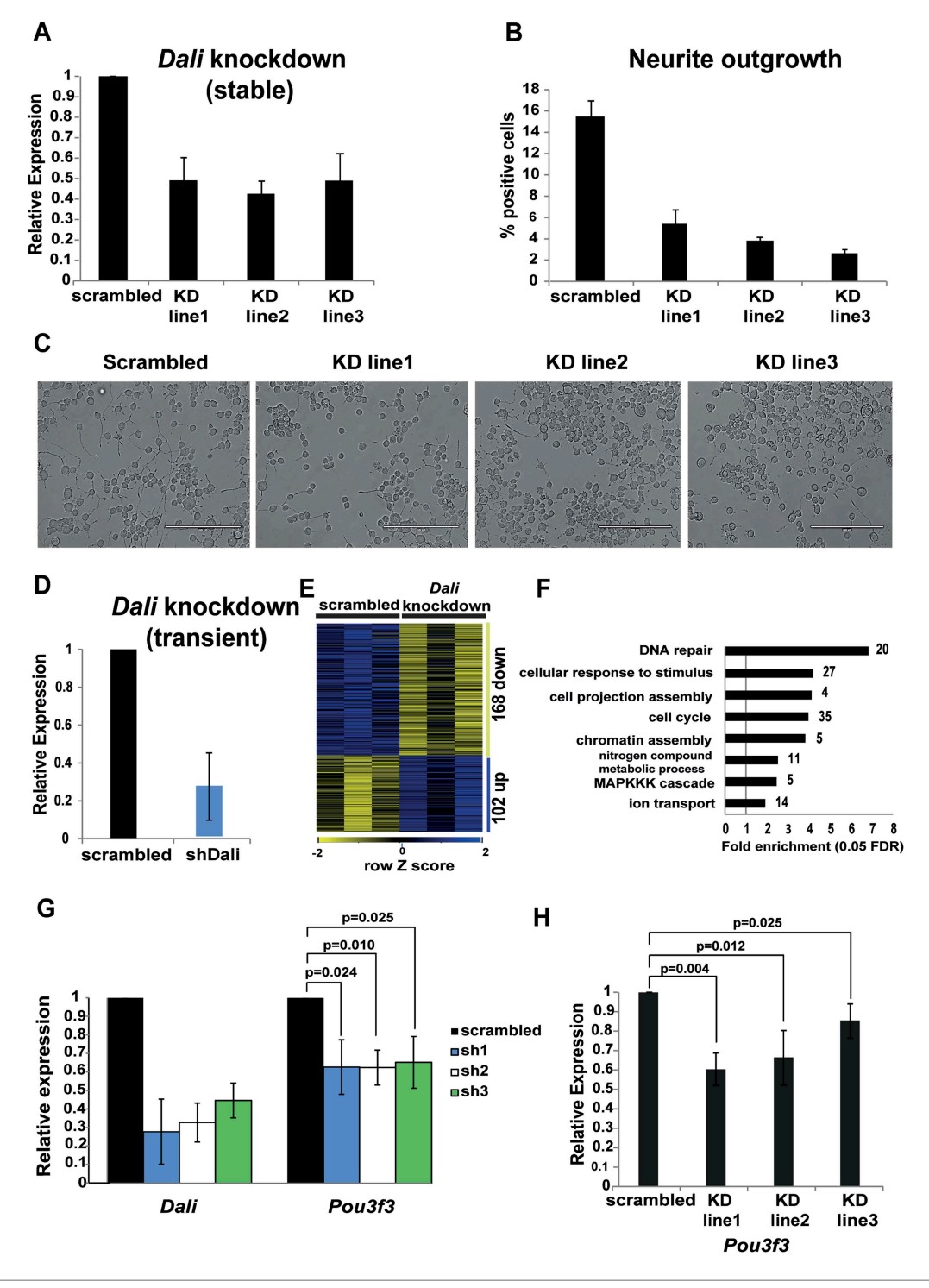

**Figure 2**. *Dali* plays a role in regulating genes in neuronal cells. (**A**) qRT-PCR analysis validates reduced levels of *Dali* in three clonal *Dali* knockdown cell lines compared to a control line. Mean values ± s.e., n = 3. (**B**) Reduced neurite outgrowth in RA-differentiated *Dali* knockdown cells. Cells were imaged using bright field microscopy. Cells with ≥1 neurites of length greater than twice the cell body diameter were scored as positive. Average values ± s.e., n = 3. 500-600 cells were counted in each case across at least three non-overlapping fields. (**C**) Representative images of control and stable *Dali* knockdown cells differentiated with RA for 72 hr. Scale bar = 200 μm. (**D**) N2A cells were transfected with either a non-targeting control (scrambled) or a *Dali* targeting shRNA expression vector
*Figure 2. Continued on next page*

*Figure 2. Continued*

(shDali) for 72 hr. Mean values ± s.e., n = 3. (**E**) Transient *Dali* knockdown induces statistically significant changes in the expression of 270 genes in N2A cells (10% FDR) (***Supplementary file 2***). (**F**) Gene Ontology (GO) categories significantly enriched among *Dali* regulated genes (5% FDR, hypergeometric test, Benjamini and Hochberg correction; ***Supplementary file 2***). (**G**) Decreased *Pou3f3* expression upon *Dali* knockdown. Normalised using *Gapdh*, shown relative to a non-targeting control (set at 1). Mean values ± s.e., n = 3, one tailed t-Test, unequal variance. (**H**) Reduced *Pou3f3* levels in stable *Dali* knockdown cells (see panel **A**). qRT-PCR results were normalised using *Gapdh* and presented relative to expression in control cells (set arbitrarily to 1). Mean values ± s.e., n = 3, one tailed t-Test, unequal variance.

The following figure supplement is available for figure 2:

**Figure supplement 1**. Non-coding transcripts in the Pou3f3 locus form a network of regulatory interactions.

## *Dali* and *Pou3f3* share transcriptional targets

To investigate whether *Dali* knockdown affects expression of the adjacent *Pou3f3* gene, we reduced its levels by transient transfection of three different shRNA constructs in N2A cells. After 72 hr, reduction of *Dali* levels by an average of 60–70% was found to reduce *Pou3f3* transcript levels by approximately 40% (***Figure 2G***). Three independent stable *Dali* knockdown clones in which *Dali* levels were reduced by 50–60% (***Figure 2A***) also showed ~15–40% lower *Pou3f3* levels (***Figure 2H***). This suggests that the *Dali* transcript positively regulates *Pou3f3* expression in an RNA-dependent manner. The genome-wide transcriptional response to *Dali* knockdown thus could be explained, in part, by its effect on *Pou3f3*.

Levels of another transcript, *AK011913*, expressed downstream of *Pou3f3* (***Figure 1A***) were reduced by approximately 55% upon *Dali* knockdown (***Figure 2—figure supplement 1A***). Reduction of *AK011913* levels by approximately 60% using shRNAs resulted in *Dali* and *Pou3f3* levels decreasing by 73% and 82%, respectively (***Figure 2—figure supplement 1B***). *Linc-Brn1a*, a lncRNA upstream of and sharing a bi-directional promoter with *Pou3f3*, was up-regulated by approximately 90% upon *AK011913* depletion (***Figure 2—figure supplement 1B***). This is reminiscent of the down-regulation of *Pou3f3* and up-regulation of *lincBrn-1a* following knockdown of another lncRNA downstream of *Pou3f3*, *lincBrn-1b* (***Figure 1A***) (***Sauvageau et al., 2013***). Together with previous reports, our data show the opposing regulatory influences of lncRNAs transcribed up- and downstream of *Pou3f3* on its expression. Non-coding transcripts expressed from the extended *Pou3f3* locus thus contribute to a complex network of regulatory interactions.

Furthermore, Chromatin Conformation Capture (3C) showed that the *Dali* promoter contacted three regions across the *Pou3f3* locus (***Figure 1A***) in ES derived neuronal precursors (***Figure 1—figure supplement 2***) : 1) an enhancer element sequence lying upstream of *Pou3f3* within the *linc-Brn1a* locus, 2) a region overlapping the 3' UTR of *Pou3f3* and full-length *AK53590* (which are both regulated by *Dali*), as well as parts of *TCONS_00000039* and *linc-Brn1b*, including a differentially methylated region reported to be important in regulating *Pou3f3* expression (***Mutai et al., 2009***), and 3) a region lying within another non-coding locus (*TCONS_00000040*) (***Ramos et al., 2013***). Neither *Dali* nor *Pou3f3* appears to play a role in initiating these DNA looping interactions because these contacts were present in E14 ES cells where neither is expressed (***Figure 1—figure supplement 2B***). Nevertheless, the *Dali* locus appears to contribute to an extended structurally and transcriptionally complex region centred on the *Pou3f3* gene.

To examine to what extent the transcriptional response to *Dali* knockdown can be explained by its effect on *Pou3f3*, we reduced the level of *Pou3f3* transcript in N2A cells by 35% using transient transfection of a *Pou3f3* targeting shRNA vector (***Figure 3A***) and using microarrays observed statistically significant expression changes in 1041 genes (FDR <10%; ***Figure 3B***). *Dali* transcript levels do not change upon *Pou3f3* depletion (***Figure 3A***). Genes differentially expressed after *Pou3f3* knockdown were enriched in categories related to cell division and cell cycle (***Figure 3C***). The intersection between the sets of genes differentially expressed in *Dali* or in *Pou3f3* knockdown cells was 6.2-fold greater than expected by chance (p-value < $2.2 \times 10^{-16}$), and represented 31% of all genes differentially expressed in *Dali* knockdown cells (***Figure 3D***). Approximately equal numbers of genes shared between the two datasets were down- (43 genes) or up-regulated (41 genes) in both experiments (***Supplementary file 3***). A strong correlation was observed between the fold-change values of differentially expressed genes

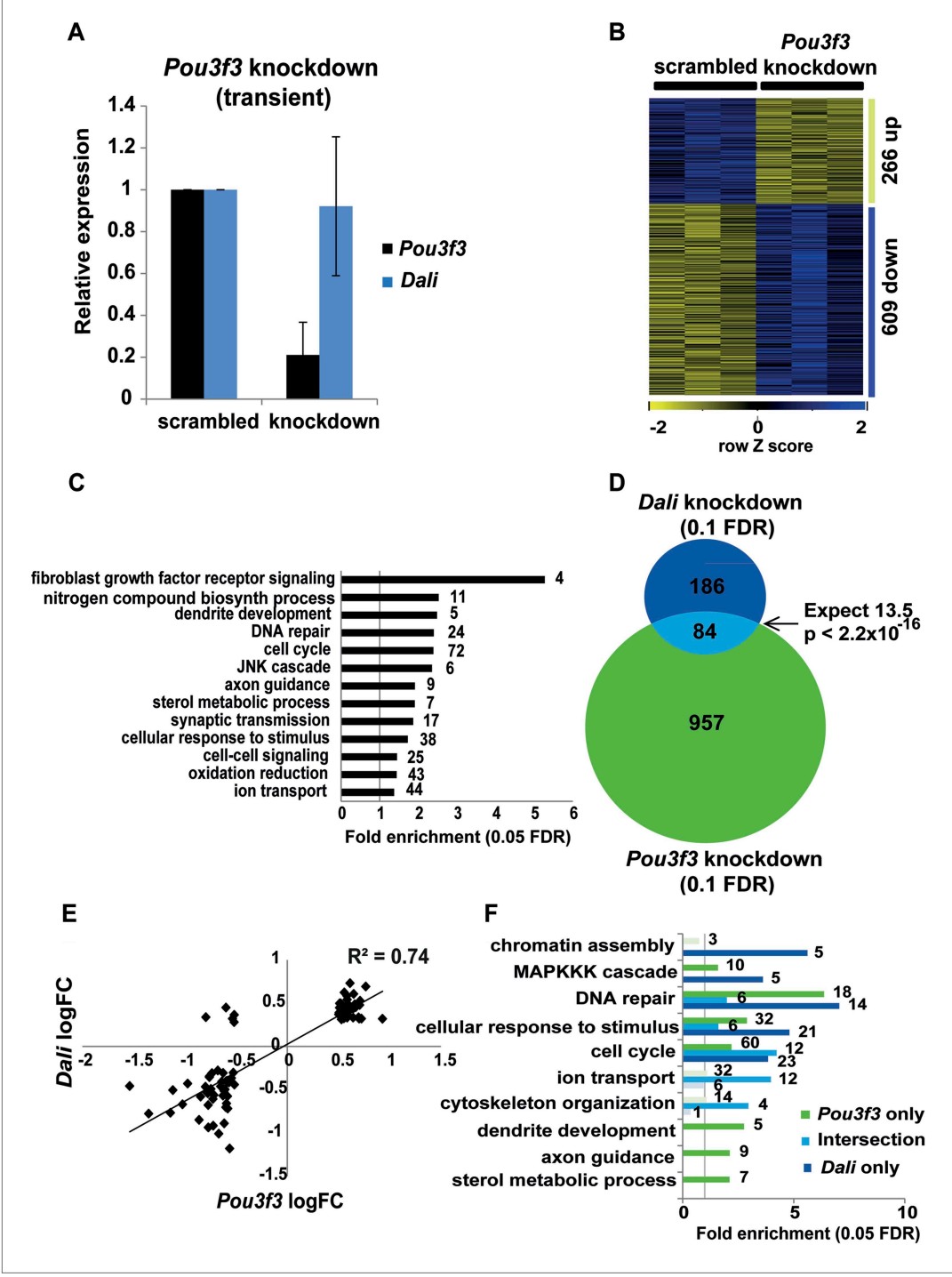

**Figure 3**. *Dali* regulates transcription in both *Pou3f3*-dependent and -independent manners. (**A**) N2A cells were transfected with either a non-targeting control (scrambled) or a *Pou3f3* targeting shRNA expression vector (knockdown). *Pou3f3* and *Dali* levels were measured by qRT-PCR 72 hr post- transfection. Mean values ± s.e., n = 3. (**B**) *Pou3f3* knockdown resulted in statistically significant changes in the expression of 1041 genes in N2A cells ((10% FDR, **Supplementary file 3**). (**C**) GO-analysis of genes differentially expressed upon *Pou3f3* analysis (5% FDR, hypergeometric test, Benjamini and Hochberg correction; **Supplementary file 3**). (**D**) Intersection of *Pou3f3* and *Dali* targets shows a significant (Fisher's exact test) overlap approximately 6.2 times as large as expected by chance alone. (**E**) Target genes common between *Dali* and *Pou3f3* show correlated expression, with the nearly all being positively or negatively regulated by both factors (R = 0.74; **Supplementary file 3**). (**F**) Enrichments of Gene Ontology categories of *Pou3f3*-dependent or -independent *Dali* targets.

in *Dali* and *Pou3f3* knockdown experiments (R = 0.74; *Figure 3E*). Genes that were significantly differentially expressed only when *Dali* was depleted were enriched in chromatin assembly and MAPKKK signalling functions, whilst genes that were differentially expressed only when *Pou3f3* transcripts were depleted were preferentially involved in dendrite development and axon guidance (*Figure 3F*). Cell cycle, DNA repair, and cellular response to stimulus genes were regulated by *Dali* in both *Pou3f3*-dependent and -independent manners. We conclude that *Dali* and *Pou3f3* interact, either genetically or molecularly, to regulate a subset of common targets involved in neural differentiation, and that *Dali* also likely possesses *Pou3f3*-independent transcriptional regulatory functions.

## *Dali* regulates gene expression programmes during neural differentiation of N2A cells

To further investigate the role of *Dali* in neuronal differentiation we profiled the transcriptomes of proliferating or RA differentiated control and *Dali* stable knockdown N2A cell lines. In proliferating cells, 733 genes were differentially expressed between *Dali* knockdown and control cells (*Figure 4A*), including many genes with functions related to neuronal differentiation, apoptosis, neuronal function (*Figure 4B*). RA-mediated neuronal differentiation induced expression changes in 958 genes in control cells and 1016 genes in *Dali* knockdown cells (*Figure 4—figure supplement 1A,B*). Based on GO category annotations, differentiation of control or *Dali* knockdown cells was broadly similar, and was associated with altered expression of cell cycle, cell differentiation, energy metabolism, and neuron projection (*Figure 4—figure supplement 1C,D*). However, 804 genes were differentially expressed between terminally differentiated control and *Dali* knockdown cells (*Figure 4C*), of which 376 genes (46.8%) also differed in expression between *Dali* knockdown and control cells prior to their differentiation (*Figure 4E*). The 428 genes that were significantly altered in expression only between stable *Dali* and control differentiated cells were enriched in functional categories relating to sterol biosynthesis, energy metabolism, cell cycle, response to chemical stimulus, cell cycle, adhesion and small GTPase signalling (*Figure 4D*). All 11 (of 34 known) sterol biosynthesis genes were down-regulated in *Dali* knockdown cells. This observation is consistent with the impaired neurite outgrowth of stable *Dali* knockdown cells because neuritogenesis and neurite outgrowth critically rely on membrane biosynthesis, and therefore, on expression of sterol biosynthesis genes (*Paoletti et al., 2011*).

In addition, several key neuronal differentiation genes, for example *Nrcam*, *Dscam*, *Dlx1* and *Pax3*, were differentially expressed between *Dali* knockdown and control cells both prior to and after differentiation. Furthermore, multifactorial analysis of RA-induced gene expression changes in control and stable *Dali* knockdown cells showed that 174 genes responded to RA differently depending on the presence or knockdown of *Dali* (FDR 5%; *Supplementary file 4*). These were significantly enriched in categories relating to neuronal development (*Figure 4F*), including pro-differentiation factors such as the inhibitor of Wnt signaling *Dkk1* (*Cajanek et al., 2009*) and Wnt receptor *Fzd5* (*Kemp et al., 2007*).

In summary, compared to control cells, stable *Dali* knockdown cells exhibit contrasting alterations in gene expression programmes before and after RA-induced differentiation. In both cases, these programmes are enriched in functional categories related to neural differentiation and function, consistent with the proposed role for *Dali* in neural development.

## *Dali* preferentially binds to active promoters in *trans*

We next sought to identify and characterise genes that are both bound and regulated by *Dali*. To do so, we determined the genome-wide binding profile of *Dali* in N2A cells using Capture Hybridisation Analysis of RNA Targets (CHART)-Seq (*Simon et al., 2011*; *Simon, 2013*) (*Figure 5—figure supplement 1A–C*). We discovered 1427 focal *Dali*-associated regions genome-wide (*Figure 5A,B*; *Supplementary file 5*), of which all nine selected loci were validated by CHART-qPCR in an independent experiment (*Figure 5—figure supplement 1D*).

*Dali* binding sites were typically limited to less than 1 kb in length (*Figure 5—figure supplement 1E*) and were distributed across the genome with no apparent chromosomal biases other than a depletion on the X chromosome which may reflect the inactivation of one X chromosome copy in these female N2A cells (*Figure 5C*). These sites were preferentially located at the 5' end of protein coding genes (*Figure 5D*): 30.5% of peaks were within 5 kb of a transcriptional start site (TSS) (*Figure 5E*). *Dali* bound sequences were significantly enriched for H3K4me3, H3K4me1 and H3K27ac modified histones and PolII occupancy, and were depleted for repressive histone marks (*Figure 5F*). This suggests that *Dali* preferentially associates with regions of active chromatin. GO category enrichment analysis

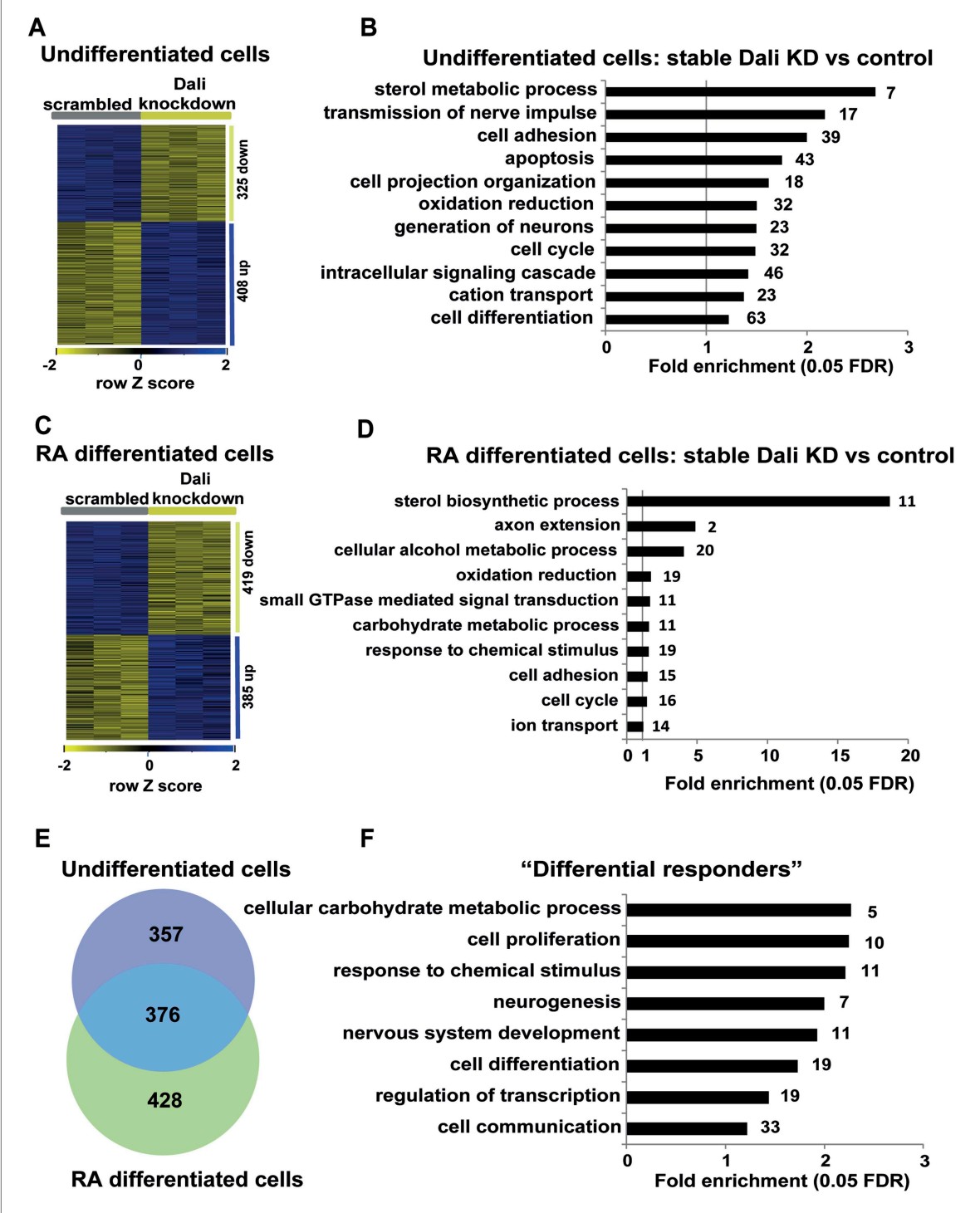

**Figure 4**. Gene expression analysis of stable *Dali* knockdown cells. (**A**) Stable *Dali* knockdown resulted in statistically significant changes in the expression of 747 genes in N2A cells (1.3-fold, 5% FDR, ***Supplementary file 4***). 332 genes were up-regulated, 415 down-regulated. (**B**) GO-analysis of genes differentially expressed upon stable *Dali* depletion (5% FDR, hypergeometric test, Benjamini and Hochberg correction). (**C**) Stable *Dali* knockdown and control cells were differentiated with retinoic acid for 72 hr. 825 genes were differentially expressed between differentiated knockdown and control lines ((≥1.3-fold, 5% FDR, ***Supplementary file 4***). 436 genes were up-regulated, 389 down-regulated. (**D**) GO-analysis of genes differentially expressed only between differentiated stable *Dali* knockdown and control cells (5% FDR, hypergeometric test, Benjamini and Hochberg correction). (**E**) Intersection of gene sets differentially expressed between stable *Dali* knockdown and control cells prior to (undifferentiated) and after retinoic acid addition
*Figure 4. Continued on next page*

# eLIFE Research article

Biochemistry | Genes and chromosomes

*Figure 4. Continued*

(differentiated). (**F**) GO-analysis of genes responding to retinoic acid treatment differently between stable *Dali* knockdown and control cells (5% FDR, hypergeometric test, Benjamini and Hochberg correction). 'Differential responder' genes were identified using multifactorial analysis of the stable *Dali* knockdown arrays using limma (***Smyth, 2004***).

The following figure supplement is available for figure 4:

**Figure supplement 1**. Transcriptomics.

showed that genes associated with *Dali* peaks contribute to processes related to neuronal differentiation (cell cycle), neuronal projection development (cytoskeleton organization and small GTPase mediated signal transduction), neuronal function (synaptic transmission), and more general cellular processes, such as gene expression, intracellular signalling, and cellular homeostasis (***Figure 5G***). 150 genes (8.6% of all *Dali* bound genes) regulated by *Dali* contained *Dali* binding sites within their regulatory regions (***Figure 5H***) and presumably represent direct transcriptional targets.

## *Dali* interacts with chromatin modifying proteins

To investigate the mechanisms of its genomic targeting, we next performed computational analysis of *Dali* bound sequences. We discovered that *Dali* binding sites do not exhibit significant sequence complementarity with the *Dali* transcript (***Figure 5—figure supplement 1F***, see Methods), and are not likely to form RNA-DNA:DNA triplex structures (***Figure 5—figure supplement 1G***), suggesting that *Dali* does not bind DNA directly. We therefore speculated that *Dali* may be targeted to the genome indirectly thorough RNA-protein interactions. To identify proteins that interact directly with *Dali*, we performed a pull down assay in which in vitro transcribed and 5′ end-biotinylated *Dali* was incubated with nuclear extract prepared from day 4 RA-differentiated ES cells. We identified, using mass spectrometry, 50 proteins that associated with *Dali*, but not with antisense *Dali* or a size-matched unrelated control transcript (***Supplementary File 7***). Direct interactions between the endogenous *Dali* transcript and four of these candidate binding proteins, the DNA methyltransferase DNMT1, the BRG1 core component of the SWI/SNF family chromatin remodelling BAF complex, and the P66beta, and SIN3A transcriptional co-factors, were subsequently validated using UV-crosslinked RNA Immunoprecipitation (UV-RIP) in N2A cells (***Figure 6A,B***). Human *DALI* was also found, using UV-RIP, to interact with human DNMT1, yet not with BRG1, in human neuroblastoma SH-SY5Y cells (***Figure 6B***). Consequently, in further experiments, we focused on the evolutionarily conserved DNMT1 interaction.

Interestingly, 9 of 58 human transcription factors reported by Hervouet et al. as interacting with the DNMT1 protein (***Hervouet et al., 2010***), including CTCF, but also AP-2, C-ets-1, LRH1, PARP, PAX6, STAT1, YY1, and Sp1, were found to have binding site motifs that were significantly enriched within our stringent *Dali* bound CHART-seq peaks (***Supplementary File 6***). Motifs for none of 42 transcription factors that do not interact with DNMT1 but interact with DNMT3a and/or DNMT3b (***Hervouet et al., 2010***) were enriched in these peaks (***Supplementary File 6***). In particular, using a de novo motif discovery approach, we found a highly-enriched CTCF-binding site-like motif in 125 out of 1427 *Dali* peaks (9%; MEME E-value = $3.1 \times 10^{-62}$; ***Figure 6C***) (***Supplementary File 7***). This result was concordant with the greater than expected overlap between *Dali*-associated regions and known CTCF binding sites in neuronal tissues (***Figure 5F***) (***Shen et al., 2012***). Using Chromatin Immunoprecipitation and qPCR (ChIP-qPCR) in N2A cells, we confirmed the CTCF-enrichment of previously-known CTCF-binding sites within 7 *Dali*-bound and regulated promoters, but not at four control regions (***Figure 6D***). However, despite CTCF and *Dali* thus occupying a subset of shared genomic binding sites, UV-RIP provided no evidence of a direct physical interaction (***Figure 6E***). Consequently, *Dali* and CTCF may be non-interacting molecular subunits of a larger ribonucleoprotein complex, or alternatively they might independently bind adjacent sequence, or compete for binding to the same region. Taken together, the data suggest that *Dali* is recruited to chromatin via indirect interactions with several DNA-binding proteins through its direct association with DNMT1.

## Depletion of *Dali* leads to DNA methylation changes at bound and regulated promoters

Increasing numbers of lncRNAs have been shown to direct DNA methylation changes at their sites of synthesis (***Mohammad et al., 2010***; ***Di Ruscio et al., 2013***). The direct interaction of *Dali* with DNMT1,

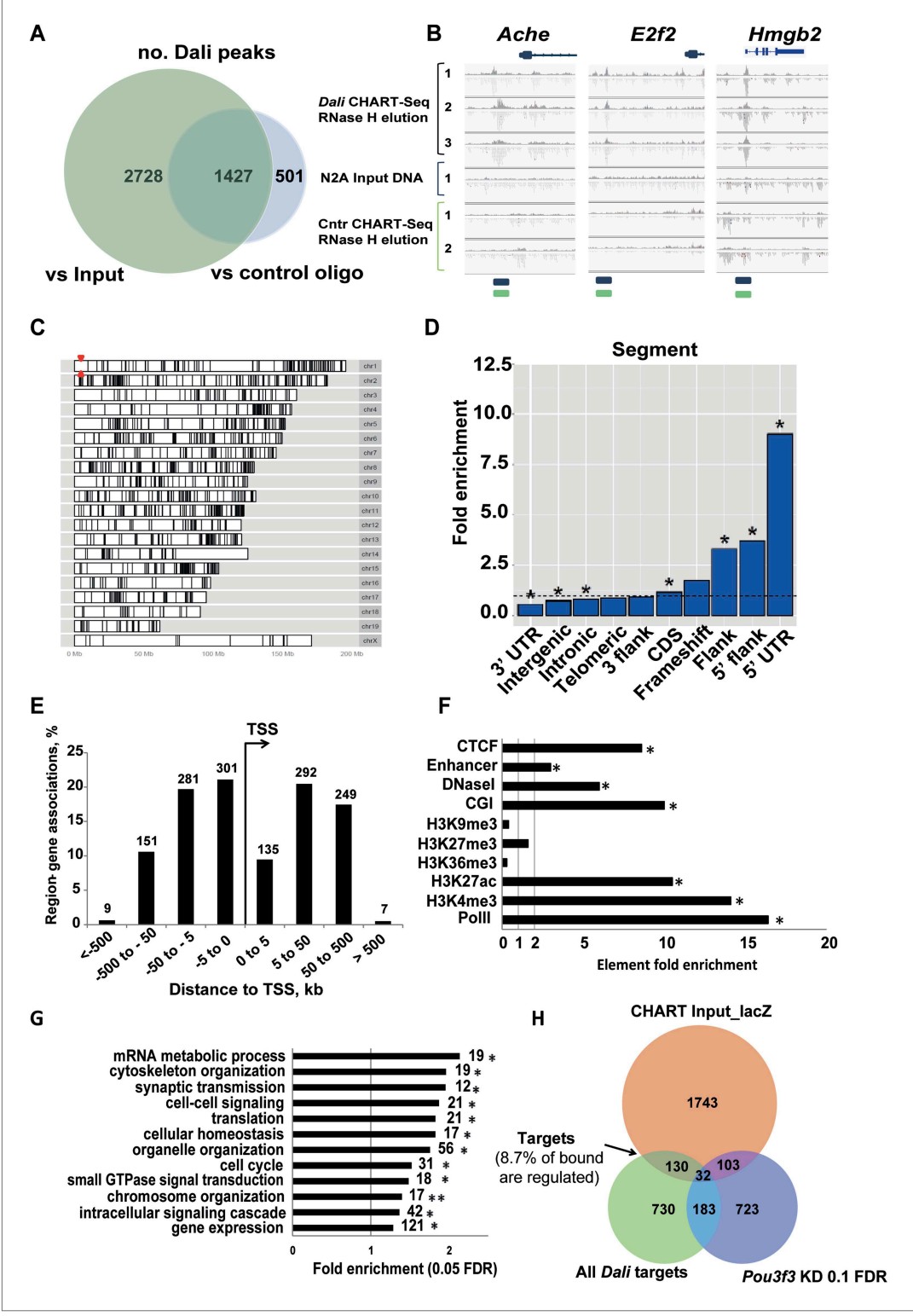

**Figure 5**. CHART-Seq analysis of *Dali* genomic binding sites. (**A**) Peaks were called against control CHART-seq experiments and input DNA. We consider only the 1427 peaks common to both comparisons (***Supplementary File 5***). (**B**) Sequencing of *Dali* bound DNA reveals focal peaks, including those at the promoter of *Ache*, *E2f2*, and *Hmgb2*. (**C** and **D**) *Dali* peaks are broadly distributed across the mouse genome (**C**) but are particularly enriched in 5′ UTRs and gene promoters (**D**). Red arrowheads in (**C**) mark the *Dali* locus. (**E**) A third of *Dali* peaks
*Figure 5. Continued on next page*

*Figure 5. Continued*

are situated within 5 kb of a TSS. (**F**) *Dali*-bound loci are enriched in active chromatin marks (H3K4me3, H3K27ac, PolII), DNase I hypersensitivity regions, enhancers and CpG islands annotations (CGI), and CTCF-bound regions, while being depleted of gene body marks (H3K36me3) and repressive chromatin marks (H3K9me3 and H3K27me3). (**G**) Representative categories from GO analysis of genes associated with *Dali* binding sites (within 1 Mb) include gene expression, cell cycle, signalling, synaptic transmission and cytoskeleton organization among others. Categories marked with an asterisk (*) are significantly enriched also among genes associated with peaks within 10 kb of a TSS, with two asterisks (**)—among genes with peaks within 100 kb (*Supplementary File 5*). (**H**) The intersection of genes proximal (<1 Mb) to *Dali* peaks, regulated by *Dali* and changing expression upon *Pou3f3* (10% FDR) knockdown identifies those both bound and regulated by *Dali*, as well as genes regulated by both *Dali* and *Pou3f3* and directly bound by *Dali*.

The following figure supplement is available for figure 5:

**Figure supplement 1**. CHART Analysis.

---

however, suggests that it may be able to regulate DNMT1-mediated CpG methylation at CpG island-associated promoters of *Dali*-bound and -regulated genes in *trans*. To investigate this, we performed Combined Bisulfite Restriction Analysis (COBRA) (*Xiong and Laird, 1997*) in parallel at 10 different CpG islands. Selection of these regions was on the basis that they each contained several COBRA-compatible restriction enzyme sites and could be efficiently amplified from bisulfite-converted template. COBRA demonstrated that five of these regions (corresponding to four genes) exhibited altered restriction profiles indicative of altered DNA methylation status after *Dali* depletion depletion (*Figure 7—figure supplement 1*). The inability of COBRA to detect changes at all sites may indicate that the DNA methylation status of the remaining regions did not change upon *Dali* depletion or that changes that occurred were undetected due to technical limitations of the assay.

Bisulfite sequencing demonstrating that the *Dlgap5*, *Hmgb2*, and *Nos1* promoters each display increased CpG methylation in two independent stable *Dali* knockdown lines compared to control further confirmed these results (*Figure 7A*). Importantly, these data show that methylation changes occur within the core of these CpG islands and are not limited to their shores. Although other unidentified factors are also likely to play a role, our results are consistent with *Dali* (or a *Dali*:POU3F3 complex) acting in *trans*, as part of a multi-subunit ribonucleoprotein complex, to reduce DNMT1-mediated CpG methylation at a subset of bound and regulated gene promoters away from its site of transcription.

One of these genes, *Nos1*, has multiple alternative promoters falling into two distinct regions (for simplicity referred to here as Exon 1 and Exon 2) whose differentiated use is proposed to fine-tune its expression in response to various physiological and developmental stimuli (*Bros et al., 2006*). Only the 5′-most region contains a CpG island and is bound by *Dali* (*Figure 7B*). By measuring expression levels of the three 5′-most *Nos1* exons in stable *Dali* knockdown and control lines we observed that the expression level of the 5′ most *Dali*-bound Exon 1 was reduced, relative to that for Exon 3, when *Dali* was depleted, whereas the expression ratio between Exons 2 and 3 was unaffected (*Figure 7C*). The preferential use of the 5′ most CpG site could reflect a secondary effect of *Dali* knockdown. Nevertheless, the observation that this site is bound by *Dali* transcript suggests that *Dali* may function by promoting the preferential use of a distantly located (and more rarely used) alternative promoter potentially through its effect on promoter-associated CpG island methylation.

## *Dali* and POU3F3 protein form a *trans*-acting transcriptional regulatory complex

A recognisable binding motif for POU III family transcription factors, such as POU3F3, was present in 115 out of 1427 *Dali* CHART-Seq peaks (8.0%; $E$-value = $3.8 \times 10^{-5}$; *Figure 6F*). This finding, together with *Dali* and *Pou3f3* regulating a set of common genes (*Figure 3D*) and *Dali* occupying regulatory regions within 135 (13%) of *Pou3f3* targets (*Figure 5H*), suggested that *Dali* and POU3F3 protein may interact physically. Indeed, we observed direct RNA-protein interactions between over-expressed FLAG-tagged POU3F3 and co-transfected *Dali*, using UV-RIP in N2A cells (*Figure 6G*). Using ChIP-qPCR, we then determined that at least five genes that were regulated by both *Dali* and *Pou3f3* contained regions that were bound both by *Dali* and by POU3F3 protein (*Figure 6H*). These results

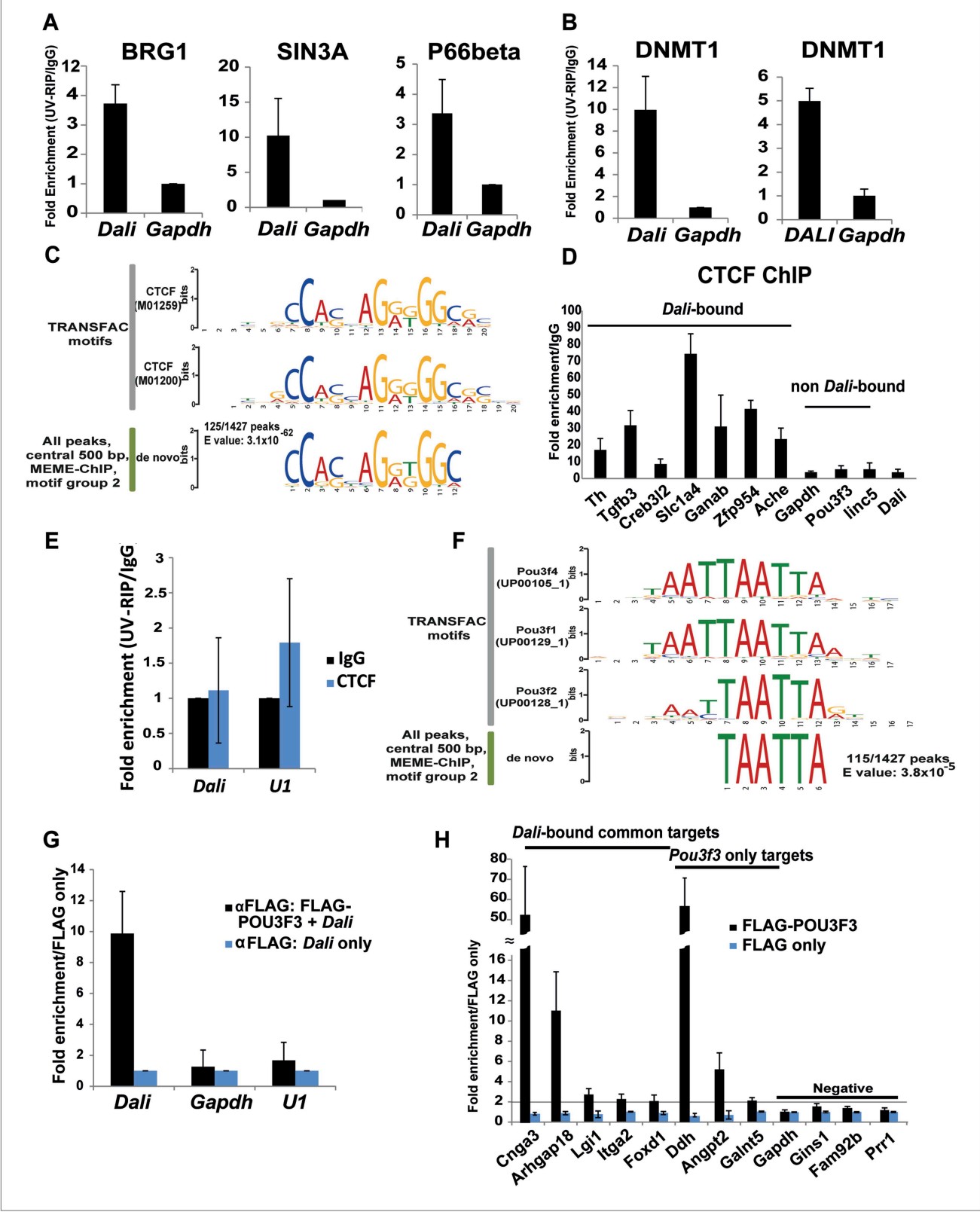

**Figure 6**. *Dali* associates with chromatin and transcriptional regulatory proteins. *Dali* interacts with BRG1, SIN3A, and P66beta in mouse N2A cells (**A**) and DNMT1 in mouse N2A and human SH-SY5Y cells (**B**). Nuclear extracts prepared from UV cross-linked cells were immuno-precipitated using either anti-DNMT1 or control IgG antibodies. Associated RNAs were purified and the levels of *Dali* and control *Gapdh* mRNA were quantified using

*Figure 6. Continued on next page*

*Figure 6. Continued*

qRT-PCR. Results are expressed as fold enrichment relative to an isotype IgG control antibody. Mean value ± s.e., n = 3. (**C**) *De novo* discovery of a near-perfect match to a CTCF motif in 125/1427 (8.8%) *Dali* CHART-Seq peaks. (**D**) *Dali* co-occupies several locations shared with CTCF. Control regions are not predicted to be bound by CTCF and are not bound by *Dali*. ChIP assays were performed in N2A cells using either an antibody against CTCF or an isotype specific control. The indicated DNA fragments were amplified using qPCR. Fold enrichment was calculated as 2-ΔΔCt (IP/IgG). Mean value ± s.e., n = 3. (**E**) *Dali* does not directly interact with CTCF protein in mouse N2A cells. Nuclear extracts were prepared from UV cross-linked cells and immuno-precipitated using either anti-CTCF or control IgG antibodies. Associated RNAs were purified and the levels of *Dali* and control *U1* snoRNA were detected in each UV-RIP using qRT-PCR. Results are expressed as fold enrichment relative to an isotype IgG control antibody. Results are presented as mean value ± s.e. of three independent experiments. (**F**) *De novo* discovery of a motif for POU III family transcription factors (which includes POU3F3) in 115/1427 (8.1%) *Dali* CHART-Seq peaks. (**G**) UV-RIP in N2A cells: FLAG-tagged POU3F3 protein directly interacts with *Dali*. Mean value ± s.e., n = 3. (**H**) ChIP-qPCR in N2A cells: POU3F3 occupies a subset of loci bound by *Dali* and regulated by both *Pou3f3* and *Dali*. Loci associated with known (*Dali*-independent) *Pou3f3* targets were used as positive control, while loci not regulated by either *Pou3f3* or *Dali* and not bound by *Dali* were used as negative control. Mean value ± s.e., n = 3.

provide further mechanistic insights into *Dali*'s mode of action and indicate that *Dali* and POU3F3 form a complex that binds to and regulates a subset of genes *in trans* in N2A cells.

## Induction of the endogenous *Dali* transcript in mouse ES cells regulates *Pou3f3* locally and *E2f2* distally

Finally, we tested whether de novo expressed *Dali* transcript can act as a transcriptional regulator in order to further substantiate the observation that *Dali* functions as a novel regulator of both local and distal gene expression. To achieve this, we induced *Dali* expression from its endogenous locus in E14 mouse ES cells, which do not express *Dali* or *Pou3f3* to detectable levels, using transient transfection of an artificial Transcription Activator-Like effector (TALE) transcription factor. After 72 hr, up-regulation of *Dali* transcript was shown to significantly increase *Pou3f3* expression (**Figure 7D**). *Dali* expression from its own locus is thus sufficient to induce the expression of its genomically neighbouring *Pou3f3* gene (**Figure 7D**). We next investigated the expression levels of *E2f2*, a gene that we found to be negatively regulated by *Dali* using shRNA mediated knockdown (**Supplementary file 2**), and found that *Dali* up-regulation reduced *E2f2* transcript levels by approximately 40% (**Figure 7D**). Taken together, these results indicate that *Dali* can regulate both local and distal target genes when its expression is induced from its endogenous locus.

## Discussion

The ability of nuclear localised lncRNAs to act *in trans* at distal genomic locations to regulate gene expression programs has been poorly understood. This is in large part because the direct transcriptional targets of only a small number of such transcripts (for example, *Paupar* (mouse), *HOTAIR*, *NEAT1*, *TERC*, *RMST* (all human), and *rox2* (*Drosophila*)) have been identified thus far (**Chu et al., 2011**; **Simon et al., 2011**; **Ng et al., 2013**; **Vance et al., 2014**). Consequently, it has been unclear how these transcripts are targeted to distal functional elements and whether thereafter they alter chromatin structure in situ.

In this study we found evidence that the intergenic lncRNA *Dali* acts both locally to regulate the expression of its nearest protein-coding gene, *Pou3f3*, and distally to regulate both *Pou3f3*-dependent and -independent target genes in an RNA-dependent manner. 8.8% (150) of all genes whose expression altered following *Dali* depletion were associated with *Dali* binding sites within 1 Mb (although 30% of peaks reside within 5 kb of a TSS, see **Figure 5E**) and, therefore, are likely to represent direct regulatory targets. This proportion lies within the range of functional sites observed for transcription factors (**Cusanovich et al., 2014**). Our results are consistent with a model in which mouse or human *Dali* is recruited to chromatin indirectly via RNA-protein interactions with both sequence-specific transcription factor proteins, such as POU3F3 which is encoded by its neighbouring gene, or non-sequence specific DNA binding cofactors including DNMT1, which in turn may interact with sequence-specific DNA-binding proteins. In this model, *Pou3f3*-dependent target genes are regulated by *Dali* both indirectly, via its transcriptional regulatory effect on the *Pou3f3* gene, and directly via its physical interaction with the POU3F3 protein and their co-occupancy at regulatory regions of target genes.

Our data show that both human and mouse *Dali* associate with DNMT1 and that depletion of *Dali* levels increases CpG methylation at *Dali* bound and regulated promoters *in trans*. Whilst a growing

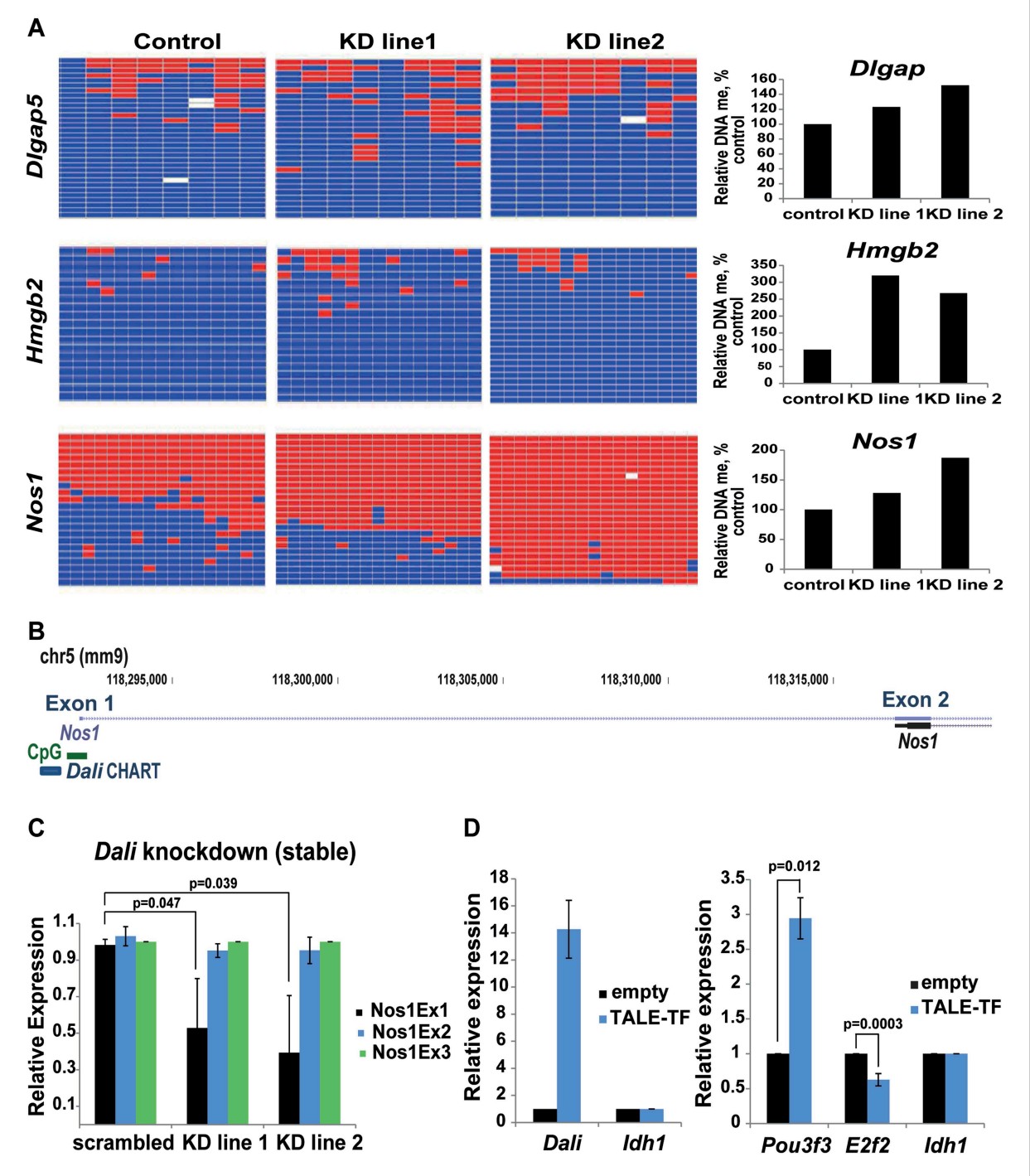

**Figure 7**. *Dali* modulates DNA methylation at bound and regulated promoters. (**A**) DNA methylation status of three CGI-associated promoters bound and regulated by *Dali* was assessed using bisulfite sequencing in control and two stable *Dali* knockdown lines. DNA methylation levels were found to be increased in knockdown lines. The degree of increase was correlated with the degree of *Dali* knockdown (see **Figure 7—figure supplement 1**). (**B**) *Nos1* gene has two clusters of alternative TSSs (Exon 1 and Exon 2). The upstream neuronal tissue-specific cluster (Exon 1) is associated with a CpG island and is bound by *Dali*. (**C**) Down-regulation of *Nos1* observed in stable *Dali* knockdown lines can be explained by reduced initiation from the *Dali*-bound TSS (Exon 1), as the ratio between Exon1 and an internal Exon 3 is diminished, while the ratio between Exon 2 and Exon 3 is not changed. Mean values ± s.e, n = 3, one tailed t-Test, unequal variance. (**D**) *Dali* transcript regulates *Pou3f3* locally and *E2f2* distally in ES mouse cells. *Dali* is expressed from its endogenous locus in non-expressing mouse E14 ES cells using custom-designed TALE-TF (left). De novo induction of the endoge-
*Figure 7. Continued on next page*

*Figure 7. Continued*

nous *Dali* locus is sufficient to up-regulate the neighbouring *Pou3f3* gene and down-regulate the distally located *E2f2* gene (right). Mean value ± s.e., n = 3.

The following figure supplement is available for figure 7:

**Figure supplement 1**. DNA Methylation analysis.

body of literature has implicated lncRNAs, such as *Kcnq1ot1* and *ecCEBPA* (***Mohammad et al., 2010***; ***Di Ruscio et al., 2013***), in modulating CpG methylation in a DNMT1-dependent manner at their sites of synthesis, our findings represent the first evidence that an intergenic lncRNA can regulate DNA methylation *in trans* at distal genomic locations away from its site of transcription.

Our findings suggest that *Dali* inhibits DNA methylation at a subset of bound and regulated regions, presumably deposited by the DNMT1 DNA methyltransferase, to which it binds. DNMT1 binds structured RNA with higher affinity than its DNA substrate (***Di Ruscio et al., 2013***). It is thus possible that *Dali* competes for binding to DNMT1 with either protein co-factors such as UHRF1, which loads and orients the enzyme on the DNA substrate (***Inomata et al., 2008***), or its DNA substrate. Targeting of DNMT1 to specific loci is believed to be mediated by DNMT1-interacting transcription factors. 58 transcriptional factors have been reported as DNMT1 interactors (***Hervouet et al., 2010***), of which 9 have enriched sequence motifs in *Dali* CHART-Seq peaks. We thus propose a model in which such transcription factors promote the sequence-specificity of *Dali*-modulated DNA methylation changes. The genomic co-localisation of DNMT1 and transcription factors using ChIP remains unknown owing to the poor performance of the available anti-DNMT1 antibodies in this application.

We have shown that *Dali* regulates genes involved in neural development and function and its depletion disrupts terminal stages of neuronal differentiation, more particularly neurite outgrowth development. *Dali* RNA binds to and up-regulates the promoters or promoter-proximal regions of key pro-differentiation factors, such as *E2f2* (***Persengiev et al., 2001***), *Fam5b* (***Terashima et al., 2010***), *Sparc* (***Bhoopathi et al., 2011***) and *Dkk1* (***Cajanek et al., 2009***) (***Watanabe et al., 2005***), as well as binding and negatively regulating genes such as *Kif2c* and *Kif11* which are known to block neurite outgrowth (***Laketa et al., 2007***; ***Myers and Baas, 2007***; ***Nadar et al., 2012***). Therefore, *Dali* works as a pro-differentiation factor in neural development by regulating the balance between proliferation and differentiation, as well as processes associated with terminal neuronal differentiation.

*Cis-* or *trans*-acting modes of action have been proposed for a growing number of lncRNAs (***Fatica and Bozzoni, 2014***). *Dali* is unusual in acting in a transcript-dependent manner to perform both local and distal gene regulatory roles like another such lncRNA, *Paupar* (***Vance et al., 2014***). *Dali* is transcribed in the vicinity of a neuronal transcription factor *Pou3f3*. Both *Dali* and *Paupar* lncRNAs are CNS-expressed and evolutionarily constrained transcripts that are co-expressed with their neighbouring transcription factor genes both spatially and temporally. Moreover, both lncRNAs interact directly with the protein product of their neighbouring genes, POU3F3 and PAX6, respectively, to regulate a large set of targets *in trans*. These observations, together with the preferential genomic location of intergenic lncRNA loci adjacent to transcription factor genes (***Ponjavic et al., 2009***) imply that lncRNAs may commonly interact with the product of genomically adjacent transcription factor genes to act *in trans* on distal genes.

## Materials and methods

### Plasmid construction

We used the Whitehead Institute siRNA selection program to design shRNAs that target multiple regions of *Dali* or *Pou3f3*. To minimise the possibility of off-target effects, we compared candidate sequences against the NCBI RefSeq database and removed those with ≥15 bases in the anti-sense strand that matched a database entry. We then cloned the double stranded DNA oligonucleotides containing sense-loop-antisense targeting sequences downstream of the U6 promoter in pBS-U6-CMVeGFP (***Sarker et al., 2005***) by linker ligation. The *Dali* expression plasmid was constructed by PCR amplifying the full length *Dali* sequence as an *Eco*RI-*Xho*I fragment from mouse N2A cell genomic DNA and inserting it into pcDNA3. The FLAG-tagged *Pou3f3* expression plasmid was

constructed by excising the full length *Pou3f3* ORF from *Pou3f3* (NM_008900) mouse cDNA clone in pCMV Entry vector (Cambridge Biosciences, UK) and inserting it into the multiple cloning site (MCS) of the N-terminal pFLAG-CMV-6a vector (Sigma–Aldrich, UK) between *Eco*RI and *Eco*RV sites. The sequences of all oligonucleotides used for cloning are shown in *Supplementary file 1*.

### *Dali* and *Pou3f3* knockdown

Cells were plated at a density of approximately $2 \times 10^5$ cells per well in a six well plate. 16–24 hr later cells were transfected with 1.5 μg shRNA expression construct using FuGENE 6 (Promega, UK) following the manufacturer's instructions. Total RNA was extracted from the cells 48–72 hr later using TRIzol-chloroform extraction method. For stable transfections, N2A cells were co-transfected with the pBSU6-shRNA expression vector and pTK-Hyg (Clontech, Mountain View, CA) at a 5:1 ratio. 72 hr post-transfection 200 μg/ml Hygromycin B was added to the cells to select individual drug resistant clones that were later isolated and expanded under selective conditions. *Dali* expression in individual clones was measured by qRT-PCR.

### qRT-PCR and RACE

Reverse transcription was performed using the QuantiTect Reverse Transcription Kit (Qiagen, Netherlands). SYBR Green quantitative PCR was performed using a Step One Plus Real-Time PCR System (Applied Biosystems, UK). For RACE, GeneRacer Kit (Invitrogen, UK) was used according to the manufacturer's instructions. Human foetal brain RNA was purchased from Promega. Primers are listed in *Supplementary file 1*.

### Cell culture

Mouse N2A neuroblastoma and E14 ES cells were cultured as described in (*Vance et al., 2014*). The N2A cell line was chosen because it has been used extensively as a model to study neural differentiation in vitro (*Shea et al., 1985*). Human neuroblastoma (SH-SY5Y) cells were grown in DMEM/F12 medium supplemented with 10% FBS, 1% penicillin-streptomycin, and 1% L-glutamine at 37°C in a humidified atmosphere with 5% $CO_2$. Biochemical fractionation, ChIP and UV-RIP experiments was performed exactly as described in *Vance et al. (2014).* The following antibodies were used: anti-DNMT1 (ab87656; Abcam, UK), anti-BRG1 (ab4081; Abcam), anti-P66beta (ab76924; Abcam), anti-SIN3A (Active Motif, Belgium, 39,865), anti-CTCF (Abcam, 70,303), anti-rabbit IgG control antibodies (Millipore, Billerica, MA) and mouse monoclonal anti-FLAG M2 beads (Sigma–Aldrich) for FLAG-tagged POU3F3 experiments.

### Animal work

All animal experiments were conducted in accordance to schedule one UK Home Office guidelines (Scientific Procedures Act, 1986). C57BL/6J, postnatal day P56 male and pregnant females were killed by cervical dislocation; whole brains were dissected in ice-cold phosphate-buffered saline (PBS) from adult (n = 2), and intrauterine stages E9 (n = 6), E10.5 (n = 6), E13.5 (n = 6), E15.5 (n = 6) and E18.5 (n = 6) mice. Brains were embedded in 5% agarose (low melting, Bioline) and sectioned using a vibrating microtome (Leica, VT1000S) into 200 μm coronal sections using a chilled solution of 1:1 mixture of RNAlater (Ambion) and PBS. Regions of interest (adult: dentate gyrus, subventricular zone and olfactory bulb; embryos: preplate, proliferative compartmenst combining ventricular and subventricular zones, and cortical plate from lateral and dorsal tiers) were dissected from individual sections using 27 gauge needles under visual guidance, using transillumination on a dissecting microscope (MZFLIII, Leica, Switzerland). Dissected samples were rinsed in RNAse free PBS/RNAlater 1:1, submerged in ice-cold RNAlater kept for 24 hr at 4°C and stored at −80°C in RNAlater until processing.

### Transcriptomic analysis

Total RNA was isolated using the Qiagen Mini RNeasy kit according to the manufacturers' instructions. RNA integrity was assessed on a BioAnalyzer (Agilent Technologies, UK). 200 ng RNA was used to produce labelled sense single stranded DNA (ssDNA) for hybridization with the Ambion WT Expression Kit, the Affymetrix WT Terminal Labelling and Controls Kit and the Affymetrix Hybridization, Wash, and Stain Kit following the manufacturer's instructions. Sense ssDNA was fragmented and the distribution of fragment lengths was assessed on a BioAnalyzer. Next, fragmented ssDNA was labelled and hybridized to the Affymetrix GeneChip Mouse Gene 1.0 ST Array (Affymetrix, UK). Arrays were processed on an Affymetrix GeneChip Fluidics Station 450 and Scanner 3000.

CEL files were analysed using the Limma, oligo, and genefilter R Bioconductor packages (*Smyth, 2004*; *Carvalho and Irizarry, 2010*). Arrays were RMA background corrected and quantile normalised. Summary expression values were calculated at the gene level. Genes whose expression changed upon *Dali* and *Pou3f3* knockdown, as well as upon retinoic acid induced differentiation of control and stable *Dali* knockdown cells, were filtered to remove genes showing little variation in expression (variance cut off of 0.5) before the identification of significant changes. In every case, the Limma Ebayes algorithm was used to identify differential expression between three knockdown and three control samples (biological replicates). 1.3-fold change cutoff was applied in every case. GOToolbox was used to perform Gene Ontology analyses ((*Martin et al., 2004*); http://genome.crg.es/GOToolBox/). Representative significantly enriched categories were selected from a hypergeometric test with a Benjamini-Hochberg corrected p-value threshold of 0.05.

## CHART

CHART Enrichment and RNase H Mapping experiments were performed as described in (*Simon, 2013*). We designed 10 biotinylated DNA capture (C)-oligos: 5 oligos complementary to the most accessible regions of *Dali*, as determined by RNase H mapping, and 5 oligos targeting the most evolutionarily conserved regions of the transcript (*Figure 5A*). These oligos were used as two cocktails of 5 oligos, and as a pool of all 10. As controls, we used an oligo designed to target the antisense *Dali* sequence (absent from the N2A transcriptome). Additionally we require peaks to not overlap with those identified in an analogous CHART-sequence experiment using the *E. coli lacZ* sequence (GSE52571) (*Vance et al., 2014*). Compared to controls, all three cocktails of *Dali* oligos showed significant enrichment of the *Dali* transcript (10-fold compared to *lacZ*), but no enrichment of the abundant mRNA *Gapdh* (*Figure 5B*). Without any prior information about *Dali* genomic binding, we considered its endogenous site of synthesis to assess the enrichment of transcript-associated DNA loci. Specific enrichment of *Dali* at its locus was observed as expected (*Figure 5—figure supplement 1*).

CHART extract was prepared from approximately $3 \times 10^8$ N2A cells per pull down and hybridized overnight with 810 pmol biotinylated oligonucleotide cocktail (*Supplementary File 1*) at room temperature with rotation. 250 µl MyOneC1 streptavidin beads (Invitrogen) were used to capture the complexes overnight at room temperature with rotation. After extensive washes, bound material was eluted using RNase H (New England Biolabs (NEB), UK) for 30 min at room temperature. Samples were treated with Proteinase K and cross-links were reversed. RNA was purified from 1/5 total sample volume using the QIAGEN miRNeasy kit. DNA was prepared from the remaining sample using the phenol:chloroform:isoamyl alcohol extraction and ethanol precipitation method. DNA was further sheared to an average fragment size of 150–300 bp using a Bioruptor (Diagenode, Belgium) and sequenced on an Illumina HiSeq (50 bp paired end).

## Computational analysis of CHART-seq data

CHART-seq was performed with three independent pull down samples (using two independent cocktails of 5 C-oligos, and one cocktail containing all 10 C-oligos) and sequenced simultaneously with a matched input sample. 50 bp, paired-end reads were mapped to the mouse genome (mm9) using bowtie with the options '–m1 –v2 –best–strata–a'. For each *Dali* sample, peaks were called against the matched N2A input sample (4208 peaks) and CHART-seq peaks previously analogously identified in N2A cells using two *lacZ* controls (1928 peaks) (*Vance et al., 2014*). Peak calls were made using the MACS2 algorithm ((*Zhang et al., 2008*); https://github.com/taoliu/MACS/blob/master/README) with the options '–mfold 10 30 –gsize = 2.39e9 –qvalue = 0.01' using the CGAT pipeline 'pipeline_mapping.py' (https://github.com/CGATOxford/cgat). Peak calls were then filtered such that only peak calls with a −log10 q value >5 were retained (FDR 0.001%).

We discovered 1427 *Dali*-associated regions genome-wide called against both input and *lacZ* control samples (*Figure 5A*; *Supplementary file 5*).

## Characterisation of *Dali* binding sites

The chromosomal distribution of *Dali* peaks was visualised using the R Bioconductor package 'ggbio' (*Yin et al., 2012*). Genome territory enrichments analysis was performed using the Genome Association Tester (GAT; (*Heger et al., 2013*)). 10,000 simulations were performed using a mappability filtered workspace and an isochore file partitioning the genome into eight bins based on regional GC content. For the chromosomal enrichment analyses, chromosomal territories were proportionally assigned to a

single virtual meta-chromosome before using GAT to test for GC and mappability corrected enrichments as above. Gene Ontology categories enriched for *Dali* binding were identified by intersecting regulatory regions for known coding genes with *Dali* binding sites. Regulatory regions for genes were defined following the GREAT definition (*McLean et al., 2010*) as a basal domain surrounding the TSS (from −5 kb to +1 kb) and extending domains upstream and downstream to the nearest gene's basal domain or to a maximum distance of 1 Mb. Enrichments were identified using GOToolbox.

*Dali* peaks were characterised using DNase I hypersensitivity (HS) data generated by the Stamatoyannopoulos lab at the University of Washington and chromatin features identified by the Ren lab at the Ludwig Institute for Cancer Research ((*Shen et al., 2012*); *ENCODE Project Consortium, 2012*). Enrichments of DNase I HS and chromatin features overlapping Dali peaks were assessed using GAT to control for mappability and regional GC content as above.

Complementarity between *Dali* sequence and binding locations was assessed using the EMBOSS Water algorithm (*Rice et al., 2000*) which performs Smith-Waterman alignment with a range of gap opening and extension penalties. RNA-DNA:DNA triplex formation was assessed using the Triplexator search software suit (*Buske et al., 2012*). The MEME-ChIP (*Machanick and Bailey, 2011*) algorithm was used to perform de novo motif discovery analysis by examining the unmasked DNA sequence of the central regions of peak locations. MEME-ChIP was run with the options '-meme-mod zoops -meme-minw 5 -meme-maxw 30–meme-nmotifs 50' using a custom background file prepared from regions flanking the peak locations using the command 'fasta-get-markov -m 2'. Enrichment of known vertebrate transcription factor binding sites from the TRANSFAC Professional database (*Matys et al., 2006*) was assessed using the AME algorithm (*McLeay and Bailey, 2010*) with the options '–method fisher–length-correct' using the sequence and background file prepared for MEME-ChIP analysis.

## 3C

E14 ES cells or day 4 ES-derived neuronal were cross-linked with 2% formaldehyde. Nuclei were prepared and permeabilized with 0.3% SDS in 1.2× restriction buffer (NEB3 for *BglII*) for 1 hr at 37°C. Then, SDS was sequestered by adding 1.8% Triton X-100. $1 \times 10^6$ nuclei (~15 μg of chromatin) were digested with 400 units of *BglII* restriction enzyme overnight, and the enzyme was inactivated. Nuclei were diluted in 1.15× T4 DNA ligation buffer (NEB), and SDS sequestered by adding 1% Triton X-100. The digested chromatin was ligated using 100 Weiss units of T4 DNA ligase for 4 hr at 16°C and treated with Proteinase K to reverse cross-links. Samples were further treated with RNase A, and DNA was phenol-chloroform extracted and ethanol precipitated.

A RP23-92N4 (CHORI; BACPAC) Bacterial Artificial Chromosome (BAC) clone covering the *Pou3f3-Dali* locus was treated as above and used as a control template for the 3C assay. Ligation products of 3C and BAC samples were quantified by qPCR. PCR reactions consisted of 300 ng 3C sample, 0.2 μM test primers and a primer corresponding to *Dali* promoter and 1× SYBR Green PCR Mastermix (Life Technologies, UK). All reactions were performed in triplicate. The mean threshold cycle (Ct) value was calculated and used to calculate relative amounts of PCR products. To normalise for different primer efficiencies, interaction frequencies were calculated by dividing the amount of PCR product obtained from the 3C sample by the amount of DNA obtained from control BAC DNA. Interaction frequencies were also normalised to *Gapdh* internal controls prepared from genomic DNA in the same manner as the BAC clone sample. All primers used are listed in *Supplementary file 1*.

## COBRA

We used COBRA to study 9 out of 44 CpG island-containing promoters bound by *Dali* and associated with genes differentially expressed between stable *Dali* knockdown and control cell lines prior to or subsequent to the RA-induced differentiation. 80–350 ng of genomic DNA was bisulfite-treated using EZ DNA Methylation Gold kit according to the manufacturer's instruction and used for PCR amplification. Primers for amplifying bisulfite converted template DNA were designed using MethPrimer software accessible at http://www.urogene.org/methprimer/ (*Li and Dahiya, 2002*). PCR products were on-column purified with QIAquick PCR Purification Kit. 250 ng to 1 μg of purified products were incubated with appropriate COBRA-compatible (*Bst*UI (NEB), *Msp*I (NEB), *Taq*I (Thermo Scientific), *Hpy*CH4IV (NEB)) or control (*Hsp*92II (Promega), *Bfa*I (NEB)) restriction enzymes overnight. Restriction products were analysed on 3% low melting point agarose gels.

## TALE-mediated up-regulation

Target regions were selected and TAL effector constructs were designed using software, tools, and information found on the TAL Effector Nucleotide Targeter 2.0 website accessible from https://tale-nt.cac.cornell.edu/. Construction of custom TALE-TFs designed to target promoter-proximal region of *Dali* to up-regulate transcription from the locus was performed as described by *Sanjana et al. (2012)*. The TALE-TF was designed to target the following region lying upstream of the TSS of *Dali*: chr1 (mm9): 42807019-42807038 ("TGTCCCTTGTCCACATATCT"). The TAL domain sequence used was as follows: NH NG HD HD HD NG NG NH NG HD HD NI HD NI NG NI NG.

## Data deposition

Microarray and CHART-Seq data have been deposited in the GEO database under accession number GSE62035 (http://www.ncbi.nlm.nih.gov/geo/query/acc.cgi?acc=GSE62035).

## Acknowledgements

We thank the High-Throughput Genomics Group at the Wellcome Trust Centre for Human Genetics for the generation of the sequencing data and OXION for use of their microarray facility. This project has been funded by the European Research Council (Project Reference 249869, DARCGENs; KWV, VC, LK), the Medical Research Council (CPP, SNS; and MRC Hub Grant G0900747 91070 for Sequencing) and the Wellcome Trust (Grant Reference 090532/Z/09/Z for Sequencing). VC is a recipient of The Darwin Trust of Edinburgh Postgraduate research Scholarship.

## Additional information

### Competing interests

CPP: Senior editor, *eLife*. The other authors declare that no competing interests exist.

### Funding

| Funder | Grant reference number | Author |
|---|---|---|
| European Research Council (ERC) | 249869, DARCGENs | Vladislava Chalei, Lesheng Kong, Keith W Vance |
| Medical Research Council (MRC) | | Stephen N Sansom, Chris P Ponting |
| The Darwin Trust of Edinburgh | Postgraduate Scholarship | Vladislava Chalei |
| European Commission (EC) | Human Brain Project | Juan F Montiel |

The funders had no role in study design, data collection and interpretation, or the decision to submit the work for publication.

### Author contributions

VC, Conception and design, Acquisition of data, Analysis and interpretation of data, Drafting or revising the article; SNS, Analysis and interpretation of data, Drafting or revising the article; LK, Analysis and interpretation of data, Contributed unpublished essential data or reagents; SL, Acquisition of data, Contributed unpublished essential data or reagents; JFM, Acquisition of data, Analysis and interpretation of data; KWV, Conception and design, Analysis and interpretation of data, Drafting or revising the article, Contributed unpublished essential data or reagents; CPP, Conception and design, Drafting or revising the article

### Author ORCIDs

Juan F Montiel, http://orcid.org/0000-0001-8919-4662
Chris P Ponting, http://orcid.org/0000-0003-0202-7816

### Ethics

Animal experimentation: All animal experiments were conducted in accordance to schedule one UK Home Office guidelines (Scientific Procedures Act, 1986).

## Additional files

### Supplementary files

• Supplementary file 1. Oligonucleotides.

• Supplementary file 2. Transient *Dali* knockdown microarray profiling.

• Supplementary file 3. Genes changing in transient *Pou3f3* knockdown and intersection with *Dali* targets.

• Supplementary file 4. Stable *Dali* knockdown gene lists.

• Supplementary file 5. CHART analysis.

• Supplementary file 6. Motif discovery.

• Supplementary File 7. Mass spectrometry identification of *Dali* associated proteins.

### Major dataset

The following dataset was generated:

| Author(s) | Year | Dataset title | Dataset ID and/or URL | Database, license, and accessibility information |
|---|---|---|---|---|
| Chalei V, Sansom SN, Kong L, Lee S, Montiel J, Vance KW, Ponting CP | 2014 | Data from: The DNMT1 associated lncRNA Dali is an epigenetic regulator of neural differentiation | GSE62035 | Publicly available at NCBI Gene Expression Omnibus. |

The following previously published dataset was used:

| Author(s) | Year | Dataset title | Dataset ID and/or URL | Database, license, and accessibility information |
|---|---|---|---|---|
| Vance KW, Sansom SN, Lee S, Chalei V, Kong L, Cooper SE, Oliver PL, Ponting CP | 2014 | The long non-coding RNA Paupar regulates the expression of both local and distal genes [CHART-seq] | http://www.ncbi.nlm.nih.gov/geo/query/acc.cgi?acc=GSE52571 | Publicly available at NCBI Gene Expression Omnibus. |

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
