## [Decision Letter]

Thank you for sending your work entitled "The DNMT1 associated lncRNA
*Dali* is an epigenetic regulator of neural differentiation" for
consideration at *eLife*. Your article has been favorably evaluated by
Detlef Weigel (Senior editor) and 2 reviewers, one of whom is a member of our Board of
Reviewing Editors. The Reviewing editor and the other reviewer discussed their comments
before we reached this decision, and the Reviewing editor has assembled the following
comments to help you prepare a revised submission.

Your work focuses on the functional roles of the lncRNA *Dali* and its
regulation not only of its neighboring transcription factor gene Pou3f3, but importantly
also at distal sites through the methylation of these sites. The topic of the functional
roles of lncRNAs is of keen interest and most importantly the mechanism of how these
roles are achieved is of critical importance. Thus, this manuscript is both timely and
important. However, there are several issues that the authors should make clearer for
the readers in order for an audience to understand the messages being communicated and
to provide a clear and supportable set of conclusions. These issues are:

1) Several places in the manuscript the authors' interpretation of their data
results in a set of specific conclusions about mechanisms or cause and effect
relationships that are not the only explanation or conclusion. Importantly, there is no
acknowledgement of other interpretations or explanations of why other interpretations
cannot be concluded. For example, while the suggestion of recruitment of
*Dali* via a complex of DNA binding proteins that include CTCF is an
interesting hypothesis, this collection of data can be interpreted by alternative
mechanisms such as the coincident binding of CTCF (due to CTCF genome-wide and abundant
binding).

2) What was the basis of choosing 10 *Dali* binding sites? What changes,
if any occur, at the other 5 sites? Given that half of the results obtained are
consistent with the model of *Dali* or *Dali:Pou3f3*
complex acting in trans to reduce DNMT1 methylation, there appears to be the likelihood
that the mechanism may minimally involve a more complex set of interactions. This should
be acknowledged.

3) “…we performed Combined Bisulfite Restriction Analysis (COBRA) (57) in parallel at 10 different
CpG islands and demonstrated that five of these regions (corresponding to 4 genes)
exhibited altered restriction profiles indicative of altered DNA methylation status
after Dali depletion […] These results are consistent with *Dali*
(or a *Dali:*POU3F3 complex, see below) acting in trans to reduce
DNMT1-mediated CpG methylation at a subset of bound and regulated gene promoters away
from its site of transcription”: this is speculation since the effect on the use
of the 5' most CpG site could be achieved through a secondary effect of the
*Dali* KD.

4) Figure 2 (heat map); are these the results of
all 3 KDs or only1? Given the results seen in Figure 2 what data indicates that some or most of these expression changes are not
explained by "off-target" effects by the sequences used in the KDs?

---

## [Author Response]

*1) Several places in the manuscript the authors' interpretation of their
data results in a set of specific conclusions about mechanisms or cause and effect
relationships that are not the only explanation or conclusion. Importantly, there is
no acknowledgement of other interpretations or explanations of why other
interpretations cannot be concluded. For example, while the suggestion of recruitment
of Dali via a complex of DNA binding proteins that include CTCF is an interesting
hypothesis, this collection of data can be interpreted by alternative mechanisms such
as the coincident binding of CTCF (due to CTCF genome-wide and abundant
binding)*.

We have now edited the main text to include additional explanations that are also
consistent with the data as suggested by the reviewers. This specific example now states
that *Dali* and CTCF “might independently bind adjacent sequence,
or compete for binding to the same region”.

*2) What was the basis of choosing 10* Dali *binding sites? What
changes, if any occur, at the other 5 sites? Given that half of the results obtained
are consistent with the model of* Dali *or* Dali:Pou3f3
*complex acting in trans to reduce DNMT1 methylation, there appears to be the
likelihood that the mechanism may minimally involve a more complex set of
interactions. This should be acknowledged.*

We selected these 10 *Dali* binding sites to maximise the likelihood of
detecting DNA methylation changes using COBRA. We ensured that each region contained
several COBRA-compatible restriction enzyme sites and could be efficiently amplified
from bisulfite-converted template.

We also now indicate that the failure to detect changes at all regions tested could
reflect either that the DNA methylation status of the remaining regions did not change
upon *Dali* depletion or that those changes that occurred were undetected
due to technical limitations of the COBRA assay.

We agree with the reviewers that *Dali* mediated DNA methylation changes
in *trans* may minimally involve a more complex set of interactions that
are in addition to *Dali*, POU3F3 and DNMT1. This is acknowledged in the
revised manuscript as suggested. We now state: “Although other unidentified
factors are also likely to play a role, our results are consistent with
*Dali* (or a *Dali*:POU3F3 complex) acting in
*trans*, as part of a multi-subunit ribonucleoprotein complex, to
reduce DNMT1-mediated CpG methylation at a subset of bound and regulated gene promoters
away from its site of transcription.”

*3) “…we performed Combined Bisulfite Restriction Analysis (COBRA)
(*[57]*) in parallel at 10 different CpG islands and
demonstrated that five of these regions (corresponding to 4 genes) exhibited altered
restriction profiles indicative of altered DNA methylation status after Dali
depletion […] These results are consistent with* Dali *(or
a* Dali*:POU3F3 complex, see below) acting in trans to reduce
DNMT1-mediated CpG methylation at a subset of bound and regulated gene promoters away
from its site of transcription”: this is speculation since the effect on the
use of the 5' most CpG site could be achieved through a secondary effect of
the* Dali *KD.*

Thank you. We now have added: “The preferential use of the 5́’ most
CpG site could reflect a secondary effect of *Dali* knockdown.
Nevertheless, the observation that this site is bound by *Dali*
transcript suggests that *Dali* may function by promoting the
preferential use of a distantly located (and more rarely used) alternative promoter
potentially through its effect on promoter-associated CpG island
methylation.”

*4)*
Figure 2
*(heat map); are these the results of all 3 KDs or only1? Given the results seen
in*
Figure 2
*what data indicates that some or most of these expression changes are not
explained by "off-target" effects by the sequences used in the KDs*?

The data in Figure 2 were generated from three
independent *Dali* knockdowns. To assess whether these expression changes
are likely to be explained by ‘off-target’ effects we used two further
shRNA expression constructs targeting different regions of the *Dali*
transcript to deplete *Dali* expression and validated expression changes
in 14 out of 15 *Dali* targets identified in the microarray. These
results are shown in Figure 2—figure supplement 1 panel C and in the modified text.